# LazySVD: Even Faster SVD Decomposition
# Yet Without Agonizing Pain*

**Zeyuan Allen-Zhu**
zeyuan@csail.mit.edu
Institute for Advanced Study
& Princeton University

**Yuanzhi Li**
yuanzhil@cs.princeton.edu
Princeton University

## Abstract

We study $k$-SVD that is to obtain the first $k$ singular vectors of a matrix $A$. Recently, a few breakthroughs have been discovered on $k$-SVD: Musco and Musco [19] proved the first gap-free convergence result using the block Krylov method, Shamir [21] discovered the first variance-reduction stochastic method, and Bhojanapalli et al. [7] provided the fastest $O(\mathsf{nnz}(A) + \mathsf{poly}(1/\varepsilon))$-time algorithm using alternating minimization.

In this paper, we put forward a new and simple LazySVD framework to improve the above breakthroughs. This framework leads to a faster gap-free method outperforming [19], and the first accelerated *and* stochastic method outperforming [21]. In the $O(\mathsf{nnz}(A) + \mathsf{poly}(1/\varepsilon))$ running-time regime, LazySVD outperforms [7] in certain parameter regimes without even using alternating minimization.

## 1 Introduction

The singular value decomposition (SVD) of a rank-$r$ matrix $A \in \mathbb{R}^{d \times n}$ corresponds to decomposing $A = V\Sigma U^\top$ where $V \in \mathbb{R}^{d \times r}$, $U \in \mathbb{R}^{n \times r}$ are two column orthonormal matrices, and $\Sigma = \mathrm{diag}\{\sigma_1, \ldots, \sigma_r\} \in \mathbb{R}^{r \times r}$ is a non-negative diagonal matrix with $\sigma_1 \geq \sigma_2 \geq \cdots \geq \sigma_r \geq 0$. The columns of $V$ (resp. $U$) are called the left (resp. right) singular vectors of $A$ and the diagonal entries of $\Sigma$ are called the singular values of $A$. SVD is one of the most fundamental tools used in machine learning, computer vision, statistics, and operations research, and is essentially equivalent to principal component analysis (PCA) up to column averaging.

A rank $k$ partial SVD, or $k$-SVD for short, is to find the top $k$ left singular vectors of $A$, or equivalently, the first $k$ columns of $V$. Denoting by $V_k \in \mathbb{R}^{d \times k}$ the first $k$ columns of $V$, and $U_k$ the first $k$ columns of $U$, one can define $A_k^* := V_k V_k^\top A = V_k \Sigma_k U_k^\top$ where $\Sigma_k = \mathrm{diag}\{\sigma_1, \ldots, \sigma_k\}$. Under this notation, $A_k^*$ is the the best rank-$k$ approximation of matrix $A$ in terms of minimizing $\|A - A_k\|$ among all rank $k$ matrices $A_k$. Here, the norm can be any Schatten-$q$ norm for $q \in [1, \infty]$, including spectral norm ($q = \infty$) and Frobenius norm ($q = 2$), therefore making $k$-SVD a very powerful tool for information retrieval, data de-noising, or even data compression.

Traditional algorithms to compute SVD essentially run in time $O(nd \min\{d, n\})$, which is usually very expensive for big-data scenarios. As for $k$-SVD, defining $\mathsf{gap} := (\sigma_k - \sigma_{k+1})/(\sigma_k)$ to be the relative $k$-th eigengap of matrix $A$, the famous subspace power method or block Krylov method [14] solves $k$-SVD in time $O(\mathsf{gap}^{-1} \cdot k \cdot \mathsf{nnz}(A) \cdot \log(1/\varepsilon))$ or $O(\mathsf{gap}^{-0.5} \cdot k \cdot \mathsf{nnz}(A) \cdot \log(1/\varepsilon))$ respectively if ignoring lower-order terms. Here, $\mathsf{nnz}(A)$ is the number of non-zero elements in matrix $A$, and the more precise running times are stated in Table 1.

Recently, there are breakthroughs to compute $k$-SVD faster, from three distinct perspectives.

| Paper | Running time  (× for being outperformed) | GF? | Stoc? | Acc? |
|---|---|---|---|---|
| subspace PM [19] | $\widetilde{O}\big(\frac{k\,\mathsf{nnz}(A)}{\varepsilon} + \frac{k^2 d}{\varepsilon}\big)$ ✗ | yes | no | no |
| | $\widetilde{O}\big(\frac{k\,\mathsf{nnz}(A)}{\mathsf{gap}} + \frac{k^2 d}{\mathsf{gap}}\big)$ ✗ | no | | |
| block Krylov [19] | $\widetilde{O}\big(\frac{k\,\mathsf{nnz}(A)}{\varepsilon^{1/2}} + \frac{k^2 d}{\varepsilon} + \frac{k^3}{\varepsilon^{3/2}}\big)$ ✗ | yes | no | yes |
| | $\widetilde{O}\big(\frac{k\,\mathsf{nnz}(A)}{\mathsf{gap}^{1/2}} + \frac{k^2 d}{\mathsf{gap}} + \frac{k^3}{\mathsf{gap}^{3/2}}\big)$ ✗ | no | | |
| LazySVD **Corollary 4.3 and 4.4** | $\widetilde{O}\big(\frac{k\,\mathsf{nnz}(A)}{\varepsilon^{1/2}} + \frac{k^2 d}{\varepsilon^{1/2}}\big)$ | yes | no | yes |
| | $\widetilde{O}\big(\frac{k\,\mathsf{nnz}(A)}{\mathsf{gap}^{1/2}} + \frac{k^2 d}{\mathsf{gap}^{1/2}}\big)$ | no | | |
| Shamir [21] | $\widetilde{O}\big(knd + \frac{k^4 d}{\sigma_k^4 \mathsf{gap}^2}\big)$    (local convergence only) ✗ | no | yes | no |
| LazySVD **Corollary 4.3 and 4.4** | $\widetilde{O}\big(knd + \frac{kn^{3/4} d}{\sigma_k^{1/2}\varepsilon^{1/2}}\big)$  $\big(\text{always } \le \widetilde{O}\big(knd + \frac{kd}{\sigma_k^2\varepsilon^2}\big)\big)$ | yes | yes | yes |
| | $\widetilde{O}\big(knd + \frac{kn^{3/4} d}{\sigma_k^{1/2}\mathsf{gap}^{1/2}}\big)$ $\big(\text{always } \le \widetilde{O}\big(knd + \frac{kd}{\sigma_k^2\mathsf{gap}^2}\big)\big)$ | no | | |
| All GF results above provide $(1+\varepsilon)\|\Delta\|_2$ spectral and $(1+\varepsilon)\|\Delta\|_F$ Frobenius guarantees | | | | |

Table 1: Performance comparison among direct methods. Define $\mathsf{gap} = (\sigma_k - \sigma_{k+1})/\sigma_k \in [0, 1]$. GF = Gap Free; Stoc = Stochastic; Acc = Accelerted. Stochastic results in this table are assuming $\|a_i\|_2 \le 1$ following (1.1).

The *first breakthrough* is the work of Musco and Musco [19] for proving a running time for $k$-SVD that does not depend on singular value gaps (or any other properties) of $A$. As highlighted in [19], providing ***gap-free*** results was an open question for decades and is a more reliable goal for practical purposes. Specifically, they proved that the block Krylov method converges in time $\widetilde{O}\big(\frac{k\,\mathsf{nnz}(A)}{\varepsilon^{1/2}} + \frac{k^2 d}{\varepsilon} + \frac{k^3}{\varepsilon^{3/2}}\big)$, where $\varepsilon$ is the multiplicative approximation error.[2]

The *second breakthrough* is the work of Shamir [21] for providing a fast ***stochastic*** $k$-SVD algorithm. In a stochastic setting, one assumes[3]

$$A \text{ is given in form } AA^\top = \frac{1}{n}\sum_{i=1}^n a_i a_i^\top \text{ and each } a_i \in \mathbb{R}^d \text{ has norm at most } 1 \ . \tag{1.1}$$

Instead of repeatedly multiplying matrix $AA^\top$ to a vector in the (subspace) power method, Shamir proposed to use a random rank-1 copy $a_i a_i^\top$ to approximate such multiplications. When equipped with very ad-hoc variance-reduction techniques, Shamir showed that the algorithm has a better (local) performance than power method (see Table 1). Unfortunately, Shamir's result is (1) not gap-free; (2) not accelerated (i.e., does not match the $\mathsf{gap}^{-0.5}$ dependence comparing to block Krylov); and (3) requires a very accurate warm-start that in principle can take a very long time to compute.

The *third breakthrough* is in obtaining running times of the form $\widetilde{O}(\mathsf{nnz}(A) + \mathsf{poly}(k, 1/\varepsilon) \cdot (n + d))$ [7, 8], see Table 2. We call them ***NNZ results***. To obtain NNZ results, one needs sub-sampling on the matrix and this incurs a poor dependence on $\varepsilon$. For this reason, the polynomial dependence on $1/\varepsilon$ is usually considered as the most important factor. In 2015, Bhojanapalli et al. [7] obtained a $1/\varepsilon^2$-rate NNZ result using alternating minimization. Since $1/\varepsilon^2$ also shows up in the sampling complexity, we believe the quadratic dependence on $\varepsilon$ is tight among NNZ types of algorithms.

All the cited results rely on ad-hoc non-convex optimization techniques together with matrix algebra, which make the final proofs complicated. Furthermore, Shamir's result [21] only works if a $1/\mathsf{poly}(d)$-accurate warm start is given, and the time needed to find a warm start is unclear.

In this paper, we develop a new algorithmic framework to solve $k$-SVD. It not only improves the aforementioned breakthroughs, but also relies only on simple convex analysis.

| Paper | Running time | Frobenius norm | Spectral norm |
|---|---|---|---|
| [8] | $O(\mathsf{nnz}(A)) + O\big(\frac{k^2}{\varepsilon^4}(n+d) + \frac{k^3}{\varepsilon^5}\big)$ | $(1+\varepsilon)\|\Delta\|_F$ | $(1+\varepsilon)\|\Delta\|_F$ |
| [7] | $O(\mathsf{nnz}(A)) + \widetilde{O}\big(\frac{k^5(\sigma_1/\sigma_k)^2}{\varepsilon^2}(n+d)\big)$ | $(1+\varepsilon)\|\Delta\|_F$ | $\|\Delta\|_2 + \varepsilon\|\Delta\|_F$ |
| LazySVD **Theorem 5.1** | $O(\mathsf{nnz}(A)) + \widetilde{O}\big(\frac{k^2(\sigma_1/\sigma_{k+1})^4}{\varepsilon^2}d\big)$ | N/A | $\|\Delta\|_2 + \varepsilon\|\Delta\|_F$ |
|  | $O(\mathsf{nnz}(A)) + \widetilde{O}\big(\frac{k^2(\sigma_1/\sigma_{k+1})^2}{\varepsilon^{2.5}}(n+d)\big)$ | N/A | $\|\Delta\|_2 + \varepsilon\|\Delta\|_F$ |
|  | $O(\mathsf{nnz}(A)) + \widetilde{O}\big(\frac{k^4(\sigma_1/\sigma_{k+1})^{4.5}}{\varepsilon^2}d\big)$ | $(1+\varepsilon)\|\Delta\|_2$ | $\|\Delta\|_2 + \varepsilon\|\Delta\|_F$ |

Table 2: Performance comparison among $O(\mathsf{nnz}(A) + \mathsf{poly}(1/\varepsilon))$ type of algorithms. Remark: we have not tried hard to improve the dependency with respect to $k$ or $(\sigma_1/\sigma_{k+1})$. See Remark 5.2.

## 1.1 Our Results and the Settlement of an Open Question

We propose to use an extremely simple framework that we call LazySVD to solve $k$-SVD:

> LazySVD: perform 1-SVD repeatedly, $k$ times in total.

More specifically, in this framework we first compute the leading singular vector $v$ of $A$, and then left-project $(I - vv^\top)A$ and repeat this procedure $k$ times. Quite surprisingly,

> *This seemingly "most-intuitive" approach was widely considered as "not a good idea."*

In textbooks and research papers, one typically states that LazySVD has a running time that inversely depends on all the intermediate singular value gaps $\sigma_1 - \sigma_2, \ldots, \sigma_k - \sigma_{k+1}$ [18, 21]. This dependence makes the algorithm useless if some singular values are close, and is even thought to be necessary [18]. For this reason, textbooks describe only block methods (such as block power method, block Krylov, alternating minimization) which find the top $k$ singular vectors together. Musco and Musco [19] stated as an *open question* to design "single-vector" methods without running time dependence on all the intermediate singular value gaps.

In this paper, we fully answer this open question with novel analyses on this LazySVD framework. In particular, the resulting running time either

- depends on $\mathsf{gap}^{-0.5}$ where $\mathsf{gap}$ is the relative singular value gap only between $\sigma_k$ and $\sigma_{k+1}$, or
- depends on $\varepsilon^{-0.5}$ where $\varepsilon$ is the approximation ratio (so is gap-free).

Such dependency matches the best known dependency for block methods.

More surprisingly, by making different choices of the 1-SVD subroutine in this LazySVD framework, we obtain multiple algorithms for different needs (see Table 1 and 2):

- If accelerated gradient descent or Lanczos algorithm is used for 1-SVD, we obtain a faster $k$-SVD algorithm than block Krylov [19].
- If a variance-reduction stochastic method is used for 1-SVD, we obtain the first accelerated stochastic algorithm for $k$-SVD, and this outperforms Shamir [21].
- If one sub-samples $A$ before applying LazySVD, the running time becomes $\widetilde{O}(\mathsf{nnz}(A) + \varepsilon^{-2}\mathsf{poly}(k) \cdot d)$. This improves upon [7] in certain (but sufficiently interesting) parameter regimes, but completely avoids the use of alternating minimization.

Finally, besides the running time advantages above, our analysis is completely based on convex optimization because 1-SVD is solvable using convex techniques. LazySVD also works when $k$ is not known to the algorithm, as opposed to block methods which need to know $k$ in advance.

**Other Related Work.** Some authors focus on the streaming or online model of 1-SVD [4, 15, 17] or $k$-SVD [3]. These algorithms are slower than offline methods. Unlike $k$-SVD, accelerated stochastic methods were previously known for 1-SVD [12, 13]. After this paper is published, LazySVD has been generalized to also solve canonical component analysis and generalized PCA by the same authors [1]. If one is only interested in projecting a vector to the top $k$-eigenspace without computing the top $k$ eigenvectors like we do in this paper, this can also be done in an accelerated manner [2].

## 2 Preliminaries

Given a matrix $A$ we denote by $\|A\|_2$ and $\|A\|_F$ respectively the spectral and Frobenius norms of $A$. For $q \geq 1$, we denote by $\|A\|_{S_q}$ the Schatten $q$-norm of $A$. We write $A \succeq B$ if $A, B$ are symmetric and $A - B$ is positive semi-definite (PSD). We denote by $\lambda_k(M)$ the $k$-th largest eigenvalue of a symmetric matrix $M$, and $\sigma_k(A)$ the $k$-th largest singular value of a rectangular matrix $A$.

Since $\lambda_k(AA^\top) = \lambda_k(A^\top A) = (\sigma_k(A))^2$,

> solving $k$-SVD for $A$ is the same as solving $k$-PCA for $M = AA^\top$.

We denote by $\sigma_1 \geq \cdots \sigma_d \geq 0$ the singular values of $A \in \mathbb{R}^{d \times n}$, by $\lambda_1 \geq \cdots \lambda_d \geq 0$ the eigenvalues of $M = AA^\top \in \mathbb{R}^{d \times d}$. (Although $A$ may have fewer than $d$ singular values for instance when $n < d$, if this happens, we append zeros.) We denote by $A_k^*$ the best rank-$k$ approximation of $A$.

We use $\perp$ to denote the orthogonal complement of a matrix. More specifically, given a column orthonormal matrix $U \in \mathbb{R}^{d \times k}$, we define $U^\perp := \{x \in \mathbb{R}^d \mid U^\top x = 0\}$. For notational simplicity, we sometimes also denote $U^\perp$ as a $d \times (d - k)$ matrix consisting of some basis of $U^\perp$.

**Theorem 2.1** (approximate matrix inverse). *Given $d \times d$ matrix $M \succeq 0$ and constants $\lambda, \delta > 0$ satisfying $\lambda I - M \succeq \delta I$, one can minimize the quadratic $f(x) := x^\top (\lambda I - M)x - b^\top x$ in order to invert $(\lambda I - M)^{-1}b$. Suppose the desired accuracy is $\|x - (\lambda I - M)^{-1}b\| \leq \varepsilon$. Then,*

- *Accelerated gradient descent (AGD) produces such an output $x$ in $O\left(\frac{\lambda^{1/2}}{\delta^{1/2}} \log \frac{\lambda}{\varepsilon\delta}\right)$ iterations, each requiring $O(d)$ time plus the time needed to multiply $M$ with a vector.*

- *If $M$ is given in the form $M = \frac{1}{n}\sum_{i=1}^n a_i a_i^\top$ and $\|a_i\|_2 \leq 1$, then accelerated SVRG (see for instance [5]) produces such an output $x$ in time $O\left(\max\{nd, \frac{n^{3/4}d\lambda^{1/4}}{\delta^{1/2}}\} \log \frac{\lambda}{\varepsilon\delta}\right)$.*

## 3 A Specific $1$-SVD Algorithm: Shift-and-Inverse Revisited

In this section, we study a specific 1-PCA algorithm `AppxPCA` (recall 1-PCA equals 1-SVD). It is a (multiplicative-)approximate algorithm for computing the leading eigenvector of a symmetric matrix.

We emphasize that, in principle, most known 1-PCA algorithms (e.g., power method, Lanczos method) are suitable for our LazySVD framework. We choose `AppxPCA` solely because it provides the maximum flexibility in obtaining all stochastic / NNZ running time results at once.

Our `AppxPCA` uses the shift-and-inverse routine [12, 13], and our pseudo-code in Algorithm 1 is a modification of Algorithm 5 that appeared in [12]. Since we need a more refined running time statement with a multiplicative error guarantee, and since the stated proof in [12] is anyways only a sketched one, we choose to carefully reprove a similar result of [12] and state the following theorem:

**Theorem 3.1** (`AppxPCA`). *Let $M \in \mathbb{R}^{d \times d}$ be a symmetric matrix with eigenvalues $1 \geq \lambda_1 \geq \cdots \geq \lambda_d \geq 0$ and corresponding eigenvectors $u_1, \ldots, u_d$. With probability at least $1 - p$, `AppxPCA` produces an output $w$ satisfying*

$$\sum_{i \in [d], \lambda_i \leq (1-\delta_\times)\lambda_1} (w^\top u_i)^2 \leq \varepsilon \quad \text{and} \quad w^\top M w \geq (1 - \delta_\times)(1 - \varepsilon)\lambda_1 \ .$$

*Furthermore, the total number of oracle calls to $\mathcal{A}$ is $O(\log(1/\delta_\times)m_1 + m_2)$, and each time we call $\mathcal{A}$ we have $\frac{\lambda^{(s)}}{\lambda_{\min}(\lambda^{(s)}I - M)} \leq \frac{12}{\delta_\times}$ and $\frac{1}{\lambda_{\min}(\lambda^{(s)}I - M)} \leq \frac{12}{\delta_\times \lambda_1}$.*

Since `AppxPCA` reduces 1-PCA to oracle calls of a matrix inversion subroutine $\mathcal{A}$, the stated conditions $\frac{\lambda^{(s)}}{\lambda_{\min}(\lambda^{(s)}I - M)} \leq \frac{12}{\delta_\times}$ and $\frac{1}{\lambda_{\min}(\lambda^{(s)}I - M)} \leq \frac{12}{\delta_\times \lambda_1}$ in Theorem 3.1, together with complexity results for matrix inversions (see Theorem 2.1), imply the following running times for `AppxPCA`:

**Corollary 3.2.**

- *If $\mathcal{A}$ is AGD, the running time of `AppxPCA` is $\widetilde{O}\left(\frac{1}{\delta_\times^{1/2}}\right)$ multiplied with $O(d)$ plus the time needed to multiply $M$ with a vector.*

- *If $M = \frac{1}{n}\sum_{i=1}^n a_i a_i^\top$ where each $\|a_i\|_2 \leq 1$, and $\mathcal{A}$ is accelerated SVRG, then the total running time of `AppxPCA` is $\widetilde{O}\left(\max\{nd, \frac{n^{3/4}d}{\lambda_1^{1/4}\delta_\times^{1/2}}\}\right)$.*

---

**Algorithm 1** AppxPCA$(\mathcal{A}, M, \delta_\times, \varepsilon, p)$        ⋄ *(only for proving our theoretical results; for practitioners, feel free to use your favorite 1-PCA algorithm such as Lanczos to replace* AppxPCA*.)*

---

**Input:** $\mathcal{A}$, an approximate matrix inversion method; $M \in \mathbb{R}^{d \times d}$, a symmetric matrix satisfying $0 \preceq M \preceq I$; $\delta_\times \in (0, 0.5]$, a multiplicative error; $\varepsilon \in (0, 1)$, a numerical accuracy parameter; and $p \in (0, 1)$, a confidence parameter.   ⋄ *running time only logarithmically depends on $1/\varepsilon$ and $1/p$.*

1:   $m_1 \leftarrow \left\lceil 4 \log\left(\frac{288 d}{p^2}\right)\right\rceil$, $m_2 \leftarrow \left\lceil \log\left(\frac{36 d}{p^2 \varepsilon}\right)\right\rceil$;
           ⋄ $m_1 = T^{\mathrm{PM}}(8, 1/32, p)$ *and* $m_2 = T^{\mathrm{PM}}(2, \varepsilon/4, p)$ *using definition in Lemma A.1*
2:   $\widetilde{\varepsilon}_1 \leftarrow \frac{1}{64 m_1}\left(\frac{\delta_\times}{6}\right)^{m_1}$ and $\widetilde{\varepsilon}_2 \leftarrow \frac{\varepsilon}{8 m_2}\left(\frac{\delta_\times}{6}\right)^{m_2}$
3:   $\widehat{w}_0 \leftarrow$ a random unit vector; $s \leftarrow 0$; $\lambda^{(0)} \leftarrow 1 + \delta_\times$;
4:   **repeat**
5:      $s \leftarrow s + 1$;
6:      **for** $t = 1$ **to** $m_1$ **do**
7:          Apply $\mathcal{A}$ to find $\widehat{w}_t$ satisfying $\left\| \widehat{w}_t - (\lambda^{(s-1)} I - M)^{-1} \widehat{w}_{t-1}\right\| \leq \widetilde{\varepsilon}_1$;
8:      $w \leftarrow \widehat{w}_{m_1} / \|\widehat{w}_{m_1}\|$;
9:      Apply $\mathcal{A}$ to find $v$ satisfying $\left\| v - (\lambda^{(s-1)} I - M)^{-1} w\right\| \leq \widetilde{\varepsilon}_1$;
10:     $\Delta^{(s)} \leftarrow \frac{1}{2} \cdot \frac{1}{w^\top v - \widetilde{\varepsilon}_1}$ and $\lambda^{(s)} \leftarrow \lambda^{(s-1)} - \frac{\Delta^{(s)}}{2}$;
11: **until** $\Delta^{(s)} \leq \frac{\delta_\times \lambda^{(s)}}{3}$
12: $f \leftarrow s$;
13: **for** $t = 1$ **to** $m_2$ **do**
14:     Apply $\mathcal{A}$ to find $\widehat{w}_t$ satisfying $\left\| \widehat{w}_t - (\lambda^{(f)} I - M)^{-1} \widehat{w}_{t-1}\right\| \leq \widetilde{\varepsilon}_2$;
15: **return** $w := \widehat{w}_{m_2} / \|\widehat{w}_{m_2}\|$.

---

**Algorithm 2** LazySVD$(\mathcal{A}, M, k, \delta_\times, \varepsilon_{\mathsf{pca}}, p)$

---

**Input:** $\mathcal{A}$, an approximate matrix inversion method; $M \in \mathbb{R}^{d \times d}$, a matrix satisfying $0 \preceq M \preceq I$; $k \in [d]$, the desired rank; $\delta_\times \in (0, 1)$, a multiplicative error; $\varepsilon_{\mathsf{pca}} \in (0, 1)$, a numerical accuracy parameter; and $p \in (0, 1)$, a confidence parameter.

1:   $M_0 \leftarrow M$ and $V_0 \leftarrow []$;
2:   **for** $s = 1$ **to** $k$ **do**
3:      $v'_s \leftarrow$ AppxPCA$(\mathcal{A}, M_{s-1}, \delta_\times/2, \varepsilon_{\mathsf{pca}}, p/k)$;
         ⋄ *to practitioners: use your favorite 1-PCA algorithm such as Lanczos to compute* $v'_s$
4:      $v_s \leftarrow \left((I - V_{s-1} V_{s-1}^\top) v'_s\right) / \left\|(I - V_{s-1} V_{s-1}^\top) v'_s\right\|$;        ⋄ *project* $v'_s$ *to* $V_{s-1}^\perp$
5:      $V_s \leftarrow [V_{s-1}, v_s]$;
6:      $M_s \leftarrow (I - v_s v_s^\top) M_{s-1} (I - v_s v_s^\top)$     ⋄ *we also have* $M_s = (I - V_s V_s^\top) M (I - V_s V_s^\top)$
7:   **end for**
8:   **return** $V_k$.

---

## 4   Main Algorithm and Theorems

Our algorithm LazySVD is stated in Algorithm 2. It starts with $M_0 = M$, and repeatedly applies $k$ times AppxPCA. In the $s$-th iteration, it computes an approximate leading eigenvector of matrix $M_{s-1}$ using AppxPCA with a multiplicative error $\delta_\times/2$, projects $M_{s-1}$ to the orthogonal space of this vector, and then calls it matrix $M_s$.

In this stated form, LazySVD finds approximately the top $k$ eigenvectors of a symmetric matrix $M \in \mathbb{R}^{d \times d}$. If $M$ is given as $M = AA^\top$, then LazySVD automatically finds the $k$-SVD of $A$.

### 4.1   Our Core Theorems

We state our approximation and running time core theorems of LazySVD below, and then provide corollaries to translate them into gap-dependent and gap-free theorems on $k$-SVD.

**Theorem 4.1** (approximation). *Let $M \in \mathbb{R}^{d \times d}$ be a symmetric matrix with eigenvalues $1 \geq \lambda_1 \geq \cdots \lambda_d \geq 0$ and corresponding eigenvectors $u_1, \ldots, u_d$. Let $k \in [d]$, let $\delta_\times, p \in (0, 1)$, and let $\varepsilon_{\mathsf{pca}} \leq$*

$\mathsf{poly}\big(\varepsilon, \delta_\times, \frac{1}{d}, \frac{\lambda_1}{\lambda_{k+1}}\big).$[4] *Then,* LazySVD *outputs a (column) orthonormal matrix* $V_k = (v_1, \ldots, v_k) \in \mathbb{R}^{d \times k}$ *which, with probability at least* $1 - p$, *satisfies all of the following properties. (Denote by* $M_k = (I - V_k V_k^\top) M (I - V_k V_k^\top)$.)

(a) *Core lemma:* $\|V_k^\top U\|_2 \leq \varepsilon$, *where* $U = (u_j, \ldots, u_d)$ *is the (column) orthonormal matrix and* $j$ *is the smallest index satisfying* $\lambda_j \leq (1 - \delta_\times)\|M_{k-1}\|_2$.

(b) *Spectral norm guarantee:* $\lambda_{k+1} \leq \|M_k\|_2 \leq \frac{\lambda_{k+1}}{1 - \delta_\times}$.

(c) *Rayleigh quotient guarantee:* $(1 - \delta_\times)\lambda_k \leq v_k^\top M v_k \leq \frac{1}{1 - \delta_\times}\lambda_k$.

(d) *Schatten-$q$ norm guarantee: for every* $q \geq 1$, *we have* $\|M_k\|_{S_q} \leq \frac{(1+\delta_\times)^2}{(1-\delta_\times)^2} \Big( \sum_{i=k+1}^d \lambda_i^q \Big)^{1/q}$.

We defer the proof of Theorem 4.1 to the full version, and we also have a section in the full version to highlight the technical ideas behind the proof. Below we state the running time of LazySVD.

**Theorem 4.2** (running time). LazySVD *can be implemented to run in time*

- $\widetilde{O}\big(\frac{k\,\mathsf{nnz}(M) + k^2 d}{\delta_\times^{1/2}}\big)$ *if* $\mathcal{A}$ *is AGD and* $M \in \mathbb{R}^{d \times d}$ *is given explicitly;*

- $\widetilde{O}\big(\frac{k\,\mathsf{nnz}(A) + k^2 d}{\delta_\times^{1/2}}\big)$ *if* $\mathcal{A}$ *is AGD and* $M$ *is given as* $M = AA^\top$ *where* $A \in \mathbb{R}^{d \times n}$; *or*

- $\widetilde{O}\big(knd + \frac{kn^{3/4}d}{\lambda_k^{1/4}\delta_\times^{1/2}}\big)$ *if* $\mathcal{A}$ *is accelerated SVRG and* $M = \frac{1}{n}\sum_{i=1}^n a_i a_i^\top$ *where each* $\|a_i\|_2 \leq 1$.

*Above, the* $\widetilde{O}$ *notation hides logarithmic factors with respect to* $k, d, 1/\delta_\times, 1/p, 1/\lambda_1, \lambda_1/\lambda_k$.

*Proof of Theorem 4.2.* We call $k$ times AppxPCA, and each time we can feed $M_{s-1} = (I - V_{s-1}V_{s-1}^\top)M(I - V_{s-1}V_{s-1}^\top)$ implicitly into AppxPCA thus the time needed to multiply $M_{s-1}$ with a $d$-dimensional vector is $O(dk + \mathsf{nnz}(M))$ or $O(dk + \mathsf{nnz}(A))$. Here, the $O(dk)$ overhead is due to the projection of a vector into $V_{s-1}^\perp$. This proves the first two running times using Corollary 3.2.

To obtain the third running time, when we compute $M_s$ from $M_{s-1}$, we explicitly project $a_i' \leftarrow (I - v_s v_s^\top)a_i$ for each vector $a_i$, and feed the new $a_1', \ldots, a_n'$ into AppxPCA. Now the running time follows from the second part of Corollary 3.2 together with the fact that $\|M_{s-1}\|_2 \geq \|M_{k-1}\|_2 \geq \lambda_k$. $\square$

## 4.2 Our Main Results for $k$-SVD

Our main theorems imply the following corollaries (proved in full version of this paper).

**Corollary 4.3** (Gap-dependent $k$-SVD). *Let* $A \in \mathbb{R}^{d \times n}$ *be a matrix with singular values* $1 \geq \sigma_1 \geq \cdots \sigma_d \geq 0$ *and the corresponding left singular vectors* $u_1, \ldots, u_d \in \mathbb{R}^d$. *Let* $\mathsf{gap} = \frac{\sigma_k - \sigma_{k+1}}{\sigma_k}$ *be the relative gap. For fixed* $\varepsilon, p > 0$, *consider the output*

$$V_k \leftarrow \texttt{LazySVD}\left( \mathcal{A}, AA^\top, k, \mathsf{gap}, O\big(\tfrac{\varepsilon^4 \cdot \mathsf{gap}^2}{k^4 (\sigma_1/\sigma_k)^4}\big), p \right) .$$

*Then, defining* $W = (u_{k+1}, \ldots, u_d)$, *we have with probability at least* $1 - p$:

$$V_k \text{ is a rank-k (column) orthonormal matrix with } \quad \|V_k^\top W\|_2 \leq \varepsilon .$$

*Our running time is* $\widetilde{O}\big(\frac{k\,\mathsf{nnz}(A) + k^2 d}{\sqrt{\mathsf{gap}}}\big)$, *or time* $\widetilde{O}\big(knd + \frac{kn^{3/4}d}{\sigma_k^{1/2}\sqrt{\mathsf{gap}}}\big)$ *in the stochastic setting (1.1).*

Above, both running times depend only poly-logarithmically on $1/\varepsilon$.

**Corollary 4.4** (Gap-free $k$-SVD). *Let* $A \in \mathbb{R}^{d \times n}$ *be a matrix with singular values* $1 \geq \sigma_1 \geq \cdots \sigma_d \geq 0$. *For fixed* $\varepsilon, p > 0$, *consider the output*

$$(v_1, \ldots, v_k) = V_k \leftarrow \texttt{LazySVD}\left( \mathcal{A}, AA^\top, k, \tfrac{\varepsilon}{3}, O\big(\tfrac{\varepsilon^6}{k^4 d^4 (\sigma_1/\sigma_{k+1})^{12}}\big), p \right) .$$

*Then, defining* $A_k = V_k V_k^\top A$ *which is a rank $k$ matrix, we have with probability at least* $1 - p$:

1. *Spectral norm guarantee:* $\|A - A_k\|_2 \leq (1 + \varepsilon)\|A - A_k^*\|_2$;

2. *Frobenius norm guarantee:* $\|A - A_k\|_F \leq (1 + \varepsilon)\|A - A_k^*\|_F$; and

3. *Rayleigh quotient guarantee:* $\forall i \in [k]$, $\left| v_i^\top A A^\top v_i - \sigma_i^2 \right| \leq \varepsilon \sigma_i^2$.

*Running time is* $\widetilde{O}\left(\frac{k\mathsf{nnz}(A) + k^2 d}{\sqrt{\varepsilon}}\right)$, *or time* $\widetilde{O}\left(knd + \frac{kn^{3/4}d}{\sigma_k^{1/2}\sqrt{\varepsilon}}\right)$ *in the stochastic setting (1.1).*

**Remark 4.5.** The spectral and Frobenius guarantees are standard. The spectral guarantee is more desirable than the Frobenius one in practice [19]. In fact, our algorithm implies for all $q \geq 1$, $\|A - A_k\|_{S_q} \leq (1 + \varepsilon)\|A - A_k^*\|_{S_q}$ where $\|\cdot\|_{S_q}$ is the Schatten-$q$ norm. Rayleigh-quotient guarantee was introduced by Musco and Musco [19] for a more refined comparison. They showed that the block Krylov method satisfies $|v_i^\top A A^\top v_i - \sigma_i^2| \leq \varepsilon \sigma_{k+1}^2$, which is slightly stronger than ours. However, these two guarantees are not much different in practice as we evidenced in experiments.

## 5 NNZ Running Time

In this section, we translate our results in the previous section into the $O(\mathsf{nnz}(A) + \mathsf{poly}(k, 1/\varepsilon)(n + d))$ running-time statements. The idea is surprisingly simple: we sample either random columns of $A$, or random entries of $A$, and then apply `LazySVD` to compute the $k$-SVD. Such translation directly gives either $1/\varepsilon^{2.5}$ results if AGD is used as the convex subroutine and either column or entry sampling is used, or a $1/\varepsilon^2$ result if accelerated SVRG and column sampling are used together.

We only informally state our theorem and defer all the details to the full paper.

**Theorem 5.1** (informal). *Let* $A \in \mathbb{R}^{d \times n}$ *be a matrix with singular values* $\sigma_1 \geq \cdots \geq \sigma_d \geq 0$. *For every* $\varepsilon \in (0, 1/2)$, *one can apply* `LazySVD` *with appropriately chosen* $\delta_\times$ *on a "carefully sub-sampled version" of* $A$. *Then, the resulting matrix* $V \in \mathbb{R}^{d \times k}$ *can satisfy*

- *spectral norm guarantee:* $\|A - VV^\top A\|_2 \leq \|A - A_k^*\|_2 + \varepsilon\|A - A_k^*\|_F$;[5]

- *Frobenius norm guarantee:* $\|A - VV^\top A\|_F \leq (1 + \varepsilon)\|A - A_k^*\|_F$.

*The total running time depends on (1) whether column or entry sampling is used, (2) which matrix inversion routine* $\mathcal{A}$ *is used, and (3) whether spectral or Frobenius guarantee is needed. We list our deduced results in Table 2 and the formal statements can be found in the full version of this paper.*

**Remark 5.2.** The main purpose of our NNZ results is to demonstrate the strength of `LazySVD` framework in terms of improving the $\varepsilon$ dependency to $1/\varepsilon^2$. Since the $1/\varepsilon^2$ rate matches sampling complexity, it is *very challenging* have an NNZ result with $1/\varepsilon^2$ dependency.[6] We have not tried hard, and believe it possible, to improve the polynomial dependence with respect to $k$ or $(\sigma_1/\sigma_{k+1})$.

## 6 Experiments

We demonstrate the practicality of our LazySVD framework, and compare it to block power method or block Krylov method. We emphasize that in theory, the best worse-cast complexity for 1-PCA is obtained by `AppxPCA` on top of accelerated SVRG. However, for the size of our chosen datasets, Lanczos method runs faster than `AppxPCA` and therefore we adopt Lanczos method as the 1-PCA method for our LazySVD framework.[7]

**Datasets.** We use datasets SNAP/amazon0302, SNAP/email-enron, and news20 that were also used by Musco and Musco [19], as well as an additional but famous dataset RCV1. The first two can be found on the SNAP website [16] and the last two can be found on the LibSVM website [11]. The four datasets give rise sparse matrices of dimensions $257570 \times 262111$, $35600 \times 16507$, $11269 \times 53975$, and $20242 \times 47236$ respectively.

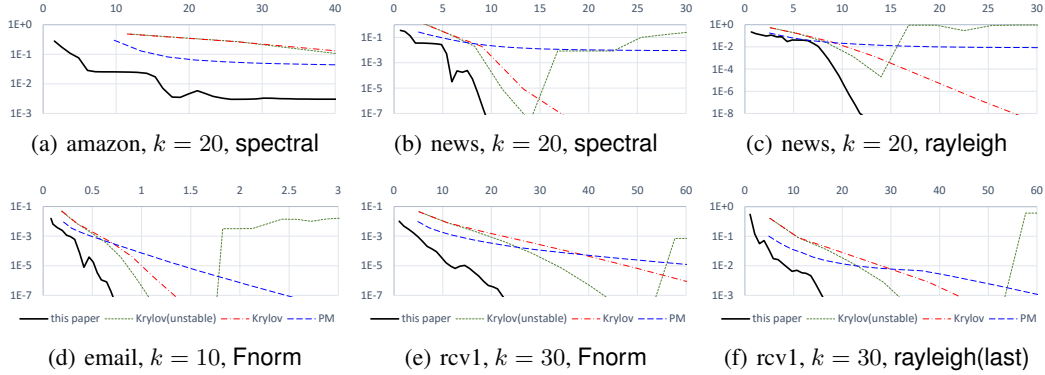

Figure 1: Selected performance plots. Relative error ($y$-axis) vs. running time ($x$-axis).

**Implemented Algorithms.** For the block Krylov method, it is a well-known issue that the Lanczos type of three-term recurrence update is numerically unstable. This is why Musco and Musco [19] only used the stable variant of block Krylov which requires an orthogonalization of each $n \times k$ matrix with respect to all previously obtained $n \times k$ matrices. This greatly improves the numerical stability albeit sacrificing running time. We implement both these algorithms. In sum, we have implemented:

- PM: block power method for $T$ iterations.
- Krylov: stable block Krylov method for $T$ iterations [19].
- Krylov(unstable): the three-term recurrence implementation of block Krylov for $T$ iterations.
- LazySVD: $k$ calls of the vanilla Lanczos method, and each call runs $T$ iterations.

**A Fair Running-Time Comparison.** For a fixed integer $T$, the four methods go through the dataset (in terms of multiplying $A$ with column vectors) the same number of times. However, since LazySVD does not need block orthogonalization (as needed in PM and Krylov) and does not need a $(Tk)$-dimensional SVD computation in the end (as needed in Krylov), the running time of LazySVD is clearly much faster for a fixed value $T$. We therefore compare the performances of the four methods in terms of running time rather than $T$.

We programmed the four algorithms using the same programming language with the same sparse-matrix implementation. We tested them single-threaded on the same Intel i7-3770 3.40GHz personal computer. As for the final low-dimensional SVD decomposition step at the end of the PM or Krylov method (which is not needed for our LazySVD), we used a third-party library that is built upon the x64 Intel Math Kernel Library so the time needed for such SVD is maximally reduced.

**Performance Metrics.** We compute four metrics on the output $V = (v_1, \ldots, v_k) \in \mathbb{R}^{n \times k}$:

- Fnorm: relative Frobenius norm error: $(\|A - VV^\top A\|_F - \|A - A_k^*\|_F)/\|A - A_k^*\|_F$.
- spectral: relative spectral norm error: $(\|A - VV^\top A\|_2 - \|A - A_k^*\|_2)/\|A - A_k^*\|_2$.
- rayleigh(last): Rayleigh quotient error relative to $\sigma_{k+1}$: $\max_{j=1}^{k} |\sigma_j^2 - v_j^\top AA^\top v_j|/\sigma_{k+1}^2$.
- rayleigh: relative Rayleigh quotient error: $\max_{j=1}^{k} |\sigma_j^2 - v_j^\top AA^\top v_j|/\sigma_j^2$.

The first three metrics were also used by Musco and Musco [19]. We added the fourth one because our theory only predicted convergence with respect to the fourth but not the third metric. However, we observe that in practice they are not much different from each other.

**Our Results.** We study four datasets each with $k = 10, 20, 30$ and with the four performance metrics, totaling $48$ plots. Due to space limitation, we only select six representative plots out of $48$ and include them in Figure 1. (The full plots can be found in Figure 2, 3, 4 and 5 in the appendix.) We make the following observations:

- LazySVD outperforms its three competitors almost universally.
- Krylov(unstable) outperforms Krylov for small value $T$; however, it is less useful for obtaining accurate solutions due to its instability. (The dotted green curves even go up if $T$ is large.)
- Subspace power method performs the slowest unsurprisingly due to its lack of acceleration.

## Footnotes

*The full version of this paper can be found on https://arxiv.org/abs/1607.03463. This paper is partially supported by a Microsoft Research Award, no. 0518584, and an NSF grant, no. CCF-1412958.

[2]In this paper, we use $\widetilde{O}$ notations to hide possible logarithmic factors on $1/\mathsf{gap}, 1/\varepsilon, n, d, k$ and potentially also on $\sigma_1/\sigma_{k+1}$.

[3]This normalization follows the tradition of stochastic $k$-SVD or 1-SVD literatures [12, 20, 21] in order to state results more cleanly.

[4]The detailed specifications of $\varepsilon_{\mathsf{pca}}$ can be found in the appendix where we restate the theorem more formally. To provide the simplest proof, we have not tightened the polynomial factors in the theoretical upper bound of $\varepsilon_{\mathsf{pca}}$ because the running time depends only logarithmic on $1/\varepsilon_{\mathsf{pca}}$.

[5]This is the best known spectral guarantee one can obtain using NNZ running time [7]. It is an open question whether the stricter $\|A - VV^\top A\|_2 \leq (1 + \varepsilon)\|A - A_k^*\|_2$ type of spectral guarantee is possible.

[6]On one hand, one can use dimension reduction such as [9] to reduce the problem size to $O(k/\varepsilon^2)$; to the best of our knowledge, it is impossible to obtain any NNZ result faster than $1/\varepsilon^3$ using solely dimension reduction. On the other hand, obtaining $1/\varepsilon^2$ dependency was the main contribution of [7]: they relied on alternating minimization but we have avoided it in our paper.

[7]Our LazySVD framework turns every 1-PCA method satisfying Theorem 3.1 (including Lanczos method) into a $k$-SVD solver. However, our theoretical results (esp. stochastic and NNZ) rely on `AppxPCA` because Lanczos is not a stochastic method.

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
