[Reviews · NeurIPS 2016]

Reviewer 1

Summary

The main result seems to be an improvement of [14] for spectral-error low rank approximation, from time k*nnz(A)/eps^{1/2} + k^2 d / eps + k^3/eps^{3/2} to k nnz(A)/eps^{1/2} + k^2 d / eps^{1/2}. This is interesting in certain regimes, if say, eps is very small and nnz(A) is about d. Note here the time is worse than nnz(A) times one can achieve with Frobenius norm error, though in certain applications one may need spectral error. My understanding of the savings - and I hope the authors can clarify this - is that [14] was a bit crude with the k/eps^{1/2} vectors in R^d they find via a block Krylov iteration. I think they just compute an orthonormal basis for such vectors which gives k^2 d/eps time (ignoring fast matrix multi optimizations). This paper achieves k^2 d / eps^{1/2} time essentially because they never have to perform such an orthogonalization; they instead iteratively find a very good rank-1 approx, then they deflate off what they just found, and repeat by finding a very good rank-1 approx on what's left, in total k times. They still need O~(1/eps^{1/2}) iterations for each rank-1 approx, and they need to deflate off what they just found, but given that the vectors they find anyway are almost already orthogonal they just pretend that they are and the deflation takes O(kd/eps^{1/2}) per iteration, giving the O(k^2 d/eps^{1/2}) in total. Another key difference with [14] is that the 1/eps^{1/2} is coming from the shift-and-invert algorithm of [7] and [8], which was not directly used in [14]. By instantiating the above framework with different algorithms in different settings, the authors obtain tradeoffs also in a stochastic setting. They also obtain nnz(A) algorithms with a mixed spectral and Frobenius guarantee though I think there is an issue here - see below.

Qualitative Assessment

I mentioned a comment in the summary above regarding intuition. I hope the authors can clarify that. Also, of importance, is for the authors to state what is the relationship with the following work which seems to use similar techniques for a similar problem: http://arxiv.org/abs/1607.02925 I didn't find the nnz(A) results very compelling. The main reason is that the authors seem to miss the earlier work of Cohen et al. "Dimensionality Reduction for k-Means Clustering and Low Rank Approximation" http://arxiv.org/pdf/1410.6801v3.pdf Applying Theorem 27 in that paper, with say, the composition of an OSNAP with an SRHT then a Gaussian, one can in nnz(A) + O~(k^2 d/eps^2) time reduce the dimension to k/eps^2 x d, at which point one can find a good low rank-k approximation of the resulting matrix using any method (this is the definition of a projection-cost preserving sketch). Importantly, the same mixed-Frobenius and spectral error guarantee is obtained. Actually the result in the Cohen et al. paper could be stronger because there is an eps/k factor in front of the Frobenius norm term, and importantly it doesn't have a sigma/sigma_{k+1} term in it, as this submission does, which can be arbitrarily large. The overall comparison of the two works depends on which algorithm one uses to solve the k/eps^2 x d problem. Just a few presentation comments: (1) the table in the stochastic setting was a bit confusing, it suggests kn^{3/4} d/(sigma_k^{1/2} eps^{1/2}) <= kd/(sigma_k^2 eps^2). which can't be true, so I guess the authors instead mean their bound is never worse than what is in parantheses. (2) in general I feel the writing style could be toned down a bit.

Confidence in this Review

2-Confident (read it all; understood it all reasonably well)


Reviewer 2

Summary

The paper proposes a new way of computing a truncated SVD, which is based on recent work (Shift-and-inverse routine). The main result concerns a gap-free convergence result, where the gap is somehow the distance between the two lasts singular values divided by their magnitude.

Qualitative Assessment

The results presented in the paper seem very impressive, and the experiments are quite convincing. I'm not able to judge the technical aspects of the paper, but, at my level, it seems to me that it's a very good paper.

Confidence in this Review

1-Less confident (might not have understood significant parts)


Reviewer 3

Summary

Provides a new iterative K-SVD algorithm with tighter accuracy/runtime guarantees

Qualitative Assessment

This paper highlights several very interesting observations. 1) pointing out that many iterative K-SVD algorithms are dependent on spectral gap, and giving a method with provable guarantees to avoid this issue. 2) proposing a vector-by-vector method that estimates each singular vector without using stochastic sampling, giving more reliable answers 3) carefully setting up the matrix inversion so that there is strong convexity, and can therefore leverage the accelerated gradient descent method's fast convergence guarantees. However, several limitations should be noted. 1) The paper claims to solve the K-SVD problem, but in fact the matrix input must be symmetric positive semidefinite with eigenvalues between 0 and 1. This is K-eig, not K-SVD. 2) After implementing the method myself, I found its results very sensitive to gap. For my naive experiment, anything with a gap > 0.01 was fine, but with a gap < 0.005 will have a very small chance of recovering the correct eigenvectors. 3) I don't think this algorithm preserves sparsity, which makes me question the scalability of it. (This is the regime where approximate K-SVD is useful, after all. Also note that both Krylov methods and subspace power methods exploit sparsity, which is one of their main selling points.) 4) The stopping condition given for the matrix inversion steps also use a matrix inverse, which is clearly impractical. 5) It would be nice if the writers provided code online. As it stands, my naive implementation runs much slower than Krylov or subspace power method. 6) A minor detail, though the writing in the main body is fine (though a bit notation dense), the title and abstract read very strangely. (For the title, the D in SVD is decomposition.)

Confidence in this Review

1-Less confident (might not have understood significant parts)


Reviewer 4

Summary

The author(s) focus on the k-SVD problem in this paper, which aims to obtain the first k singular vectors of a matrix. Specifically, they improve the recent studies by presenting a new framework which can be characterized in three aspects, i.e., a faster gap-free convergence speed, the first accelerated and stochastic method, and better parameter regimes without using alternating minimization.

Qualitative Assessment

The paper studies the popular k-SVD problem by proposing a simple framework to find the singular vectors for k iterations. The paper presents Table 1 to clearly illustrate the performance comparison among the proposed algorithm and other baselines. Overall, it is easy to capture the contributions of the paper, and the proposed framework seems effective and interesting. Here are some questions and suggestions. 1. The paper should be re-organized by providing more detailed verification of the theorems, e.g., Theorem 3.1, Theorem 4.1, Corollary 4.4, Theorem 5.1. It is a little difficult to understand the theorem in the current version, although the ideas of their proofs are presented. 2. It is confused to me that in the beginning of experiments, the author(s) mention that Lanczos method is adopted as a replacement of the proposed method AppxPCA because of the faster speed of Lanczos method. In previous section, AppxPCA is set to be an important component of IterSVD. So this replacement seems contrary to the purpose of the experiments, which aims to verify the practicality of the proposed framework. 3. More analysis and reasoning should be given about the observations of the results at the end of the experiments.

Confidence in this Review

2-Confident (read it all; understood it all reasonably well)


Reviewer 5

Summary

The authors propose a algorithm called ItrSVD and show an improved bounds of solving classic SVD problem. The algorithm use the recent advance [7] as the building block to solve each eigenvector one by one, which makes this work less novel. Also, the author actually implement Lanczos as the building block in the experiment. So I treat the whole algorithm as the Lanczos algorithm for solving k-SVD but with better theoretical analysis. However, the main contribution of this work is to answer how to decide the precision of solving each eigenvector, which is related to the convergence behavior and show an improved bound. The experimental results seem promising but there are some missing parts.

Qualitative Assessment

My main concern would be the experiment results. Although the theoretical analysis is based on [7], the authors used Lanczos algorithm as the building block in the experiments. I had some experience about using alternative minimization (ALS) an Lanczos algorithm on some computer vision applications. My previous experience show ALS is faster than Lanczos in practice. Therefore, I would like to see the comparison on these two algorithm during the rebuttal and more implementation details (some code if possible). Minor comments: 1. For a fair comparison, the authors use single thread, which is acceptable. However, for the practical use, I would like to know which algorithm is faster if we are allowed to use more threads. Will power method still be the fastest one? 2. The authors mention the constant and the polynomial dependency can be further improved. It would be appreciated if the authors provide some scratches or ideas.

Confidence in this Review

2-Confident (read it all; understood it all reasonably well)


Reviewer 6

Summary

The paper describes a new algorithmic framework to compute k-SVD of a matrix A via computing singular vectors sequentially rather than together. The authors prove that their computations are time-independent of the relative differences between the singular values of A (gap free) and that their framework outperforms the the current state of the art in many cases. In particular, they obtain faster accelerated as well as accelerated and stochastic algorithms than previously documented in literature.

Qualitative Assessment

It would improve readability if theorems (theorems 2.1, 3.1, corollaries 4.4, 4.5 for example) are preceded with some brief exposition summarizing or highlighting their content and relation to the later technical developments in the paper.

Confidence in this Review

1-Less confident (might not have understood significant parts)